# iTRAQ-Based Proteomic and Physiological Analyses Reveal the Mechanisms of Dehydration and Cryopreservation Tolerance of *Sophora tonkinensis* Gagnep. Seeds

**DOI:** 10.3390/plants12091842

**Published:** 2023-04-29

**Authors:** Yongjian Luo, Yixin Zhang, Yu Jiang, Zhangyan Dai, Qing Li, Jiaolin Mou, Li Xu, Shiming Deng, Jitao Li, Ru Wang, Jun Liu, Zhijun Deng

**Affiliations:** 1Hubei Key Laboratory of Biologic Resources Protection and Utilization, Hubei Minzu University, Enshi 445000, China; 202030361@hbmzu.edu.cn (Y.L.); 202030234@hbmzu.edu.cn (Y.J.); moujiaolin@hbmzu.edu.cn (J.M.); 2022025@hbmzu.edu.cn (L.X.); dengshiming@hbmzu.edu.cn (S.D.); ljtyouth@foxmail.com (J.L.); 202030230@hbmzu.edu.cn (R.W.); 2Agro-biological Gene Research Center, Guangdong Academy of Agricultural Sciences, Guangzhou 510640, China; zhangyixin@agrogene.ac.cn (Y.Z.); daizhangyan@agrogene.ac.cn (Z.D.); liqing@agrogene.ac.cn (Q.L.); 3Guangdong Key Laboratory for Crop Germplasm Resources Preservation and Utilization, Guangzhou 510640, China; 4Research Center for Germplasm Engineering of Characteristic Plant Resources in Enshi Prefecture, Hubei Minzu University, Enshi 445000, China; 5The Plant Germplasm Resources Laboratory, School of Forestry and Horticulture, Hubei Minzu University, Enshi 445000, China

**Keywords:** DEP, germination, LEA, orthodox seeds, reactive oxygen species (ROS), HSP

## Abstract

*Sophora tonkinensi* is a shrub of the genus *Sophora* in the family Fabaceae with anti-inflammatory and pain-relieving effects. While the cultivation, chemical makeup, and medicinal properties of *S. tonkinensis* have been reported, the physiological mechanisms governing its dehydration and cryopreservation tolerance of seeds remain unclear. In this study, we investigated the morphological, physiological, biochemical, and protein expression characteristics of *S. tonkinensis* seeds subjected to dehydration and cryopreservation techniques via the observation of cell microstructure, determination of antioxidant enzyme activity, and iTRAQ-based proteomic analysis, respectively. The results of the study demonstrated that the seeds possessed a certain level of tolerance to dehydration. The highest germination percentage of 83.0% was observed after 2 h of dehydration (10.1% water content), which was identified as the optimal time point for cryopreservation. However, the germination percentage was reduced to only 30.5% when the water content reached 5.4%, indicating that *S. tonkinensis* seeds exhibit intermediate storage behavior. Further investigation revealed that during seed dehydration and cryopreservation treatment, liposomes were gradually and highly fused, whereas the activities of ROS scavenging and stress defense were significantly enhanced. During dehydration, the seed tissues formed a protective mechanism of stress resistance based on protein processing in the endoplasmic reticulum and antioxidant system, which was related to the dehydration tolerance. Moreover, only three differentially expressed LEA proteins were identified, and it is speculated that the strengthening of intracellular metabolism and the absence of specific LEA and dehydrins could be crucial factors for the reduced germination percentage after excessive dehydration and cryopreservation.

## 1. Introduction

Dehydration tolerance (DT) is defined as the ability of an organism or tissue to survive without irreversible damage in the state of losing all or almost all of the cellular water [1]. DT of seeds is an adaptive mechanism for plants to ensure the survival and reproduction of species in the long-term evolution process and plays a key role in the preservation of plant seeds and germplasm resources [2]. It was found that the DT of orthodox seeds was gradually formed during the development process and reached the peak at physiological maturity stage. Orthodox seeds could maintain vitality after losing up to 95% of water, and then gradually lost DT during the stage of germination; recalcitrant seeds were intolerant to dehydration and highly sensitive to water loss throughout development; intermediate seeds had some tolerance to dehydration [3].

Seeds DT is a complex trait. A series of processes or mechanisms have been suggested to confer or contribute to DT, including accumulation of putatively protective molecules, e.g., late embryogenesis abundant (LEA) proteins and dehydrins [4], presence and efficient operation of antioxidant system [5], intracellular de-differentiation [6], metabolic ‘switching off’ [7], the presence and operation of repair systems during the rehydration of seeds [8,9,10], protective effects of heat shock proteins (HSP) and non-reducing raffinose families [11,12,13,14], and transcription factor networks and regulation of the AFL (ABA INSENSITIVE3, FUSCA3, and LEAFY COTYLEDON 2) subfamily [2,15,16]. The coordinated induction of protective mechanisms involves complex regulatory networks and signaling pathways. The degree of DT of seeds and embryos was negatively correlated with the respiratory activity of mitochondria, which was higher in recalcitrant seeds than in normal ones and the respiratory activity of recalcitrant seeds was higher than that of orthodox seeds [17]. During dehydration, the H_2_O_2_ content, the production rate of superoxide anion radical (O_2_^·−^), and the content of thiobarbituric acid reactive substance of DT embryos (axes) were significantly lower than that of dehydration-sensitive embryos (axes) [18], while the activity of reactive oxygen species (ROS) scavenging systems, including enzymatic and non-enzymatic, was significantly higher. During seed maturation, the accumulation of LEA proteins, small HSPs, and non-reducing oligosaccharides was closely related to the formation of seeds DT [2]. The AFL subfamily of B3 transcription factors increased the seeds (embryo) DT through positively regulating the storage substances and protective proteins accumulation [19,20]. In seeds, the parallel ABA and DOG1 (DELAY OF GERMINATION 1) signaling pathways activate the synthesis of raffinose family oligosaccharides, the expression of LEA genes and HSP genes, thereby regulating the initiation of DT and transition to dormancy [21,22,23]. In addition, seed dehydration and development were also regulated by epigenetic pathways. For example, seed desiccation in the late mature stage could also trigger nuclear DNA methylation [15,24], which was usually related to long-term repression of gene transcription [25]. Michalak et al. reported the effect of desiccation on orthodox seeds of *Pyrus communis* L., with an immediate increase in overall DNA methylation levels at seed maturity [24]. In addition, non-coding RNAs, PTMS of histones such as acetylation, methylation, and ubiquitination, played important roles in seed formation and dehydration [25,26,27]. In contrast to the remarkable progress made in the mechanisms of DT during orthodox seeds development, our understanding of intermediate and recalcitrant seeds’ desiccation-sensitivity remains limited.

*Sophora tonkinensi* was a shrub in the *Sophora* genus, belonging to the Fabaceae family, mainly in southwest China but also in northern Vietnam [28]. The matrine alkaloids in *S. tonkinensis* had the effect of anti-inflammatory and analgesic, which were the main ingredients of medicine [29,30,31]. A large number of studies has been conducted on identification, cultivation, chemical composition, and pharmacology of *S. tonkinensis* [32,33,34,35,36]. However, there is a lack of information regarding the preservation and reproduction of its seeds. The wild resources of *S. tonkinensis* were narrowly distributed, with a low natural reproduction rate coupled with indiscriminate mining, making the wild resources decrease year by year, and were close to endangered status, which made the study of seed preservation and seed characteristics critical [37]. Previous studies have shown that seed propagation was the main means of cultivating *S. tonkinensis*, using the method of sowing as it was harvested. The seeds of *S. tonkinensis* are considered to be intermediate seeds due to difficulty in long-term storage and reduced germination percentage after dehydration. In this study, we aimed to investigate the response mechanism of *S. tonkinensis* seeds to dehydration using cell microscopic observation, antioxidant analysis, and differentially expressed protein analysis and to explore the effect of cryopreservation treatment on seed DT and vigor. It provides a theoretical foundation for the safe preservation of resources of intermediate seeds.

## 2. Results

### 2.1. Dehydration Tolerance of S. tonkinensis Seeds

The initial water content of freshly matured *S. tonkinensis* seeds was as high as 31%, which were dehydrated rapidly in the active silica gel at first, dropping to 13.0% after 1 h-dehydration, and then the water content gradually slowed down until to 5.4% after 24 h-dehydration (Figure 1A). The slight degree of dehydration of *S. tonkinensis* seeds, i.e., the water content, was 10.1~13%, promoted germination, with a highest germination percentage of 84%, indicating that fresh mature *S. tonkinensis* seeds have a certain degree of DT. As the degree of dehydration increased, i.e., the water content was 5.4~9.7%, the germination percentage decreased, and the germination percentage was only 30.5% when the water content was 5.4% (Figure 1B).

The water content during the cryopreservation process affected the germination percentage of seeds. All freshly harvested seeds, with 31.0% water content, died after cryopreservation, while the germination percentage after cryopreservation was 67%, 75%, 77%, and 83% at water content between 8.5% and 13.0%, respectively, all of which were not significantly different from that before cryopreservation treatment. Among which, the highest germination percentage was 83.0% for the seeds treated with cryopreservation after dehydration to a water content of 10.1% (Figure 1B). The above results indicated that *S. tonkinensis* seeds were intermediate seeds. After appropriate rapid dehydration treatment, the seeds have a fairly high seed vigor and certain DT, which laid foundation for cryopreservation of *S. tonkinensis* seeds. 

### 2.2. Effects of Dehydration and Cryopreservation after Dehydration Treatment on the Cell Structure of S. tonkinensis Seed Plumular Axis

When the *S. tonkinensis* seeds were not treated with dehydration, the plumular axis cell structure was clear; the nucleus, nucleolus, and nuclear membrane were clearly visible; the nucleolus was dense; and the nucleoplasm was well-defined and clearly visible, showing a good structural and functional state. The shape of lipid bodies was regular, the cytoplasm was thick, the degree of vacuolization was low, and there were few mitochondria (Figure 2A). After 2 h-dehydration, irregular plasmolysis was observed in the plumular axis cells, the nuclear membrane was wrinkled, and the size of the cell contents such as lipid bodies and protein bodies did not differ significantly from that of the non-dehydrated cells (Figure 2B). After 24 h-dehydration, air bubbles appeared in the cell wall; the nucleus was in a regular spherical shape; the lipid bodies were highly fused; the nuclear membrane shrank inward; the structure of the mitochondrial was disordered; and the lipid bodies were broken and deformed (Figure 2D). After 2 h-dehydration, the seeds were stored in liquid nitrogen for 24 h. It was observed that (Figure 2C), compared to pre-cryopreservation (Figure 2B), the space between the cells was bigger; the protein bodies were evenly distributed around the cell wall; the nuclear membrane and cell membrane shrunk more severely; the nucleolus was reduced and regular spherical shape; and the lipid bodies were highly fused. After 24 h-dehydration whereafter by cryopreservation, the cell wall collapsed severely; vacuoles appeared between the cells; the lipid bodies ruptured and deformed; the cell wall ruptured; contents overflowed; and the structures such as the endoplasmic reticulum could not be identified (Figure 2E).

Combined with Figure 2A,B,D, it can be seen that, after dehydration treatment, a large number of small lipid bodies fused into larger cellulites, and cryopreservation treatment made the fusion degree of lipid bodies higher, according to combinations Figure 2B,C and Figure 2D,E.

### 2.3. Response of Antioxidant Enzyme Activities in S. tonkinensis Seed Plumular Axis Treated with Dehydration and Dehydration Followed by Cryopreservation Treatment

The superoxide dismutase (SOD) enzyme activity in *S. tonkinensis* seed plumular axis showed an overall increasing trend with the dehydration time increased, reaching maximum at 2 or 8 h-dehydration. The SOD activity after cryopreservation reached maximum at 2 or 24 h-dehydration, and the difference was not significant before and after cryopreservation at 2 h-dehydration (Figure 3A). Catalase (CAT) enzyme was the most important antioxidant enzyme responsible for scavenging excess H_2_O_2_ in cells. After dehydration and dehydration followed by cryopreservation, the CAT activity showed an increasing trend with increasing dehydration time. It is worth noting that the CAT activity was greatly increased after 2h-dehydration followed by cryopreservation. After 24h-dehydration followed by cryopreservation, the CAT activity was significantly reduced, which was consistent with the trend of seed germination percentage (Figure 3B). The activity of the APX enzyme in the plumular axis of *S. tonkinensis* seed exhibited an initial increase and subsequently decreased with increasing dehydration time. However, the overall trend increased after the dehydration followed by cryopreservation treatment. The APX activity had a peak at 2 h-dehydration, and there was no significant difference before and after cryopreservation treatment. It was noted that, after 24 h-dehydration, APX enzyme treated with cryopreservation increased rapidly, contrary to the trend of CAT activity (Figure 3C).

### 2.4. Results of iTRAQ Analysis and Differential Protein Screening of S. tonkinensis Seeds Plumular Axis Treated with Dehydration and Dehydration Followed by Cryopreservation Treatment

To investigate the molecular response pathways of *S. tonkinensis* seeds plumular axis treated with dehydration and cryopreservation, all samples were analyzed by iTRAQ proteomics. The iTRAQ proteomics produced a total of 978,057 spectra from the 4 sets of samples. We identified 97,996 known spectra, 66,953 unique spectra, 12,530 peptides, 9568 unique peptides, and 3898 proteins, respectively (Appendix A). Based on previous studies [38], differentially expressed proteins (DEPs) were defined based on a 1.2–1.5 fold change threshold. Among the proteins, only one protein in two or three replicates with fold changes ≥1.2 was defined up-regulated or ≤0.83 as down-regulated. Based on above criteria, 41, 45, 98, 40 DEPs were identified in D2 vs. CK, CD2 vs. CK, D24 vs. CK, and CD24 vs. CK, respectively (Figure 4A; Appendix A), using the control group as reference to compare the DEPs of samples. Among them, there were 7 DEPs in all comparison groups, and there were largest number of unique DEPs with 48 in D24 vs. CK (Figure 4B).

### 2.5. KEGG and Go Functional Annotation of DEPs in S. tonkinensis Seeds Axis Treated with Dehydration and Cryopreservation

GO (Gene Ontology) and KEGG (Kyoto Encyclopedia of Genes and Genomes) enrichment analyses were performed on the DEPs to investigate functions and biological pathways during dehydration and post-dehydration cryopreservation treatment. GO functional enrichment analysis showed that the DEPs before and after preservation cryopreservation significantly enriched in the structural molecule activity were structural molecule activity (GO:0005198), intracellular (GO:0005622), gene expression(GO:0010467) and intracellular ribonucleoprotein complex (GO:0030529)(Figure 5B,D). Notably, GO:0016209 antioxidant activity was significantly enriched in 2 and 24 h-dehydration but not in cryopreservation (Figure 5A–D, Appendix A). To gain insight into the metabolic pathways of DEPs, scatter plots were used to show the top 10 KEGG enrichment pathways, which showed that DEPs were significantly enriched in ribosome (ko03010), carbon fixation in photosynthetic organisms (ko00710), and alpha-linolenic acid metabolism (ko00592) pathways after 2 h-dehydration (Figure 5E, D2 vs. CK). Compared with 2 h-dehydration, the DEPs of 24 h-dehydration were significantly enriched not only in ribosome (ko03010) but also in glutathione metabolism, protein processing in endoplasmic reticulum (ko04141), and spliceosome pathways (Figure 5G, D24 vs. CK). In 2 h-dehydration followed by cryopreservation seeds, DEPs were mainly enriched in ribosome (ko03010) and protein processing in endoplasmic reticulum (ko04141) (Figure 5F, CD2 vs. CK). DEPs of 24 h-dehydration followed by cryopreservation seeds were mainly enriched in ribosome (ko03010), proteasome, and flavonoid Biosynthesis (Figure 5H, CD24 vs. CK). The physiological processes of dehydration and post-dehydration cryopreservation treatment were very similar and were both involved in ribosome, protein processing in endoplasmic reticulum, antioxidant, and signal transduction activities.

To investigate the relationship between identified key proteins, 140 DEPs were predicted using Protein–Protein Interaction Networks. The STRING online software was used to analyze the protein interaction regulatory network; among them, 68 DEPs interacted with each other (Figure 6), and the other proteins formed independent branches. GO functional analysis showed that the molecular functions of DEPs were mainly enriched in structural molecule activity, antioxidant activity, carbon–sulfur lyase activity, endopeptidase activity, intramolecular transferase activity, transferring amino groups, oligosaccharyl transferase activity, and peroxidase activity pathways and involved in the biological process of macromolecule metabolic process and response to stress (Appendix A). KEGG functional annotation showed that the DEPs were mainly involved in proteasome, protein processing in endoplasmic reticulum, RNA translation, oxidation-reductive process, and ribosome, etc. (Figure 6). However, the proteins involved in energy metabolism were not included in the network. PPI network analysis suggested that redox balance, nucleic acid metabolism, protein processing, and modification process played important roles during dehydration and cryopreservation. Interestingly, dehydration-related proteins such as LEA4, which was down-regulated in this network, and its homologous proteins SMPs (seed maturation proteins, PM41, LEA4), which were up-regulated at 2 h-dehydration and had no difference at 24 h-dehydration, were found to form an independent network (Figure 6).

## 3. Discussion

### 3.1. Molecular Mechanism of Dehydration Tolerance in S. tonkinensis Seeds

According to their storage behavior, seeds were classified into three types: orthodox seeds, intermediate seeds, and recalcitrant seeds [39,40]. DT and low temperature tolerance are the key characteristics of non-orthodox seeds, including recalcitrant seeds and intermediate seeds, to achieve long-term safe storage, which has important theoretical and practical significance. In this study, the fresh and mature *S. tonkinensis* seeds had a water content of 31% and demonstrated a certain degree of dehydration tolerance. Even after being dehydrated to 10.1% water content, the germination percentage remained at 83.0%. However, upon further dehydration to 5.4%, the germination percentage reduced to 30.5% (Figure 1). Although *S. tonkinensis* seeds could survive under relatively low water content, they could not tolerate more water loss as orthodox seeds. This storage behavior was similar to other intermediate seeds such as *Carica papaya* L. and *Pistia stratiotes* L., etc. Therefore, according to the existing definition of storage behavior [41], *S. tonkinensis seeds* should be classified as intermediate seeds.

The proteomics results showed that the expression of phosphoinositide phospholipase C was up-regulated both by dehydration and cryopreservation treatments. The electron microscopy sections revealed that there was a large number of lipid bodies that fused into large cellulites after dehydration treatment in plumular axis, and the degree of fusion increased with longer dehydration time. During cryopreservation, the medium outside the seed cells froze, whilst the cells have not yet frozen. With the increasing extracellular ice crystals, the cells lost water and gradually became dehydrated due to the vapor pressure difference. In this study, the degree of lipid bodies fusion was increased with cryopreservation treatment under the same dehydration time (Figure 2). It was reported that that oil seeds had higher DT compared to other seeds, and that high lipid body content was the most significant feature of oil seeds [42,43,44,45]. For example, rape seeds could maintain seed vigor when the water content was reduced to less than 1.5% [46]. Therefore, it is speculated that the presence of a large number of lipid bodies in the plumular axis is one of the important reasons for a certain degree of dehydration tolerance in *S. tonkinensis* seeds.

The activity of ROS scavenging system of seeds was closely related to seed recalcitrance production [47,48,49]. The activity of antioxidant enzymes of *S. tonkinensis* seeds was enhanced after 0~8 h-dehydration and cryopreservation (Figure 3), indicating that plant defense system generated self-help measures to enhance the activity of antioxidant enzymes to fight against ROS free radicals produced by dehydration and low temperature stress. SOD activity during dehydration and post-dehydration cryopreservation treatment of *S. tonkinensis* seeds gradually increased, consistent with the results of the study on dehydration tolerance of *Citrus wilsnii* Tanaka seeds [50]. Cryopreservation treatment exerted the greatest promoting effect on APX enzyme activity, and the longer the dehydration time was, the higher the APX activity was, similar to the pattern of *Sanionia uncinata* (Hedw.) Loeske reported by Pizarro et.al. [51]. The proteomic results showed that the expression of antioxidant enzymes such as peroxidase (POD), glutathione peroxidase (GPX), glutathione S-transferase (GST), CAT, and SOD were up-regulated during dehydration and post-dehydration cryopreservation (Appendix A), similar to the change trend of enzyme activities. On the one hand, it suggested the reliability of proteomic results. On the other hand, it provided a speculation that the expression and enzyme activities of antioxidant enzymes were increased to fight and scavenge ROS free radicals during dehydration and post-dehydration cryopreservation treatment, which may be another important reason for DT of *S. tonkinensis* seeds.

In this study, the protein processing in the endoplasmic reticulum pathway showed different enriched degrees in multiple treated samples (Figure 5). In this pathway, the S-Phase Kinase Associated Protein 1 (SKP1) was down-regulated at all stages, while the rest of the proteins were up-regulated at different stages. The endoplasmic reticulum played an important role in the folding, assembly, transport, secretion, and glycosylation of protein synthesis [52]. In the normal course of growth, molecular chaperones and glycosylation modifications assisted proteins in folding correctly [53]. Once stimulated, a large number of misfolded and unfolded proteins were produced and accumulated, resulting in endoplasmic reticulum stress (ERS) due to the inability of the precise endoplasmic reticulum quality control system (ERQC) system to process in time, which in turn triggered an unfolded protein response (UPR), and if a series of regulated downstream genes still failed to restore the plant endoplasmic reticulum homeostasis, UPR would initiate programmed cell death [52]. HSP was an important anti-stress-protective protein of protein processing in the endoplasmic reticulum pathway, with molecular chaperone properties, and maintained the normal function through various biological processes such as protein synthesis and degradation, stabilization of protein structure, and cellular localization [54]. In this study, three HSPs were found to be up-regulated during 2 h-dehydration followed cryopreservation treatment, and one HSP was up-regulated in all treatments (Appendix A). Notably, three PDIs (protein disulfide-isomerase) were up-regulated after 24 h-dehydration in this pathway, and studies have shown that up-regulation of PDI protein expression increased seed storage tolerance during development and abiotic stress [55,56,57], suggesting that PDI may increase storage tolerance by removing the wrong proteins in the endoplasmic reticulum, thereby regulating seed DT and low temperature tolerance [58]. In addition, the thioredoxin-like superfamily protein precursor was up-regulated in all treatments. The redox state of disulfide bonds played an important role in maintaining the correct folding and advanced structure of proteins, and excessive ROS would affect the equilibrium state of disulfide bonds between proteins, leading to structural changes and inactivation of proteins [59]. Thioredoxin (TRX) contained redox-active disulfide bonds, which could regulate redox in cells through reversible conversion of thiol-disulfide bonds and stabilize the structure and function of proteins [60]. It is hypothesized that *S. tonkinensis* seeds are subject to protein misfolding under stress conditions during dehydration, and the identified DEPs of endoplasmic reticulum-associated proteins are involved in the repair of misfolded protein macromolecules. This could potentially be another important factor for DT of *S. tonkinensis* seeds.

A reduction in metabolic activity was a characteristic of DT [61]. In this study, it was observed that DEPs from all samples were significantly enriched in the ribosomal pathway (Figure 6), and the expression of ribosomal proteins was up-regulated (Appendix A). Additionally, a group of up-regulated DEPs with molecular functions related to carbon-sulfur lyase activity, endopeptidase activity, proteasome, RNA translation, intramolecular transferase activity, and other GO terms were identified (Figure 5A–D). It is speculated that high metabolic activity could be one of the important factors contributing to the intermediate storage behavior of *S. tonkinensis* seeds.

The accumulation of LEA proteins and non-reducing raffinose family oligosaccharides was closely related to DT formation [2]. During the dehydration stage, LEAs proteins form a protective aqueous membrane around the internal cell structure and macromolecules, thus making the seeds DT due to their hydrophilic quality [62]. The 126, 123, and 167 LEA proteins were identified in normal seeds of *Arachis hypogaea* L., *Arachis hypogaea* L., and *Triticum aestivum* L., respectively [63,64,65]. Li et al. identified 50 LEA proteins in *Linum usitatissimum* L., of which 36 LuLEA proteins were expressed in seed maturation [66]. Shen et al. identified 20 LEA proteins in lotus embryo, of which 7 protein families were identified in the dehydration maturation stage [67]. At the same time, Zhang et al. identified 12 LEA proteins by proteomics, 9 of which belong to the LEA protein family [68]. However, non-orthodox seeds were different. For example, 39 LEA genes were identified in the genome of tea tree, and only 6 *CsLEA* genes were involved in seed dehydration, and most of them were repressed [69]. Julien et al. compared the results of LEA polypeptide assays during dehydration in orthodox seeds of *Medicago truncatula* Gaertn. and recalcitrant seeds of *Castanospermum australe* A. Cunn. et Fraser cotyledons and found that transcripts of 16 homologues out of 17 LEA genes for which polypeptides were detected in orthodox seeds. For the 12 LEA genes, however, the polypeptide was either absent or strongly reduced in recalcitrant seeds [70]. In this study, only 17 LEA protein polypeptide segments were identified, belonging to the 5 subfamilies of LEA2, SMP, dehydrin, LEA4, and LEA5. However, only two homologous SMP proteins were up-regulated during 2 h-dehydration, and one Lea4 protein was down-regulated during seed dehydration and post-dehydration cryopreservation (Figure 6). The above results indicate that the number of LEA proteins and families differed greatly between orthodox and non-orthodox seeds [26]. During the dehydration process, a large amount of oligosaccharides and non-reducing sugars were accumulated to maintain the stability of membranes and proteins [71]. However, KEGG functional annotation showed that the proteins involved in the pathway of glucose metabolism were not included in the network (Figure 6), and there were no changes in the synthesis of proteins related to oligosaccharides. It is speculated that the lack of sufficient LEA, dehydrin, and oligosaccharides is one of the important factors for the intermediateness of *S. tonkinensis* seeds.

### 3.2. Cryopreservation Treatment May Be Used as a Candidate Method for the Preservation of S. tonkinensis Seeds

So far, recalcitrant seeds have not been effectively preserved on a large scale for a long time, and cryopreservation was considered to be the most promising research direction [72,73]. The DT and low temperature tolerance of intermediate seeds were between orthodox seeds and recalcitrant seeds, with limited DT, while dehydration treatment did not prolong storage life of intermediate seeds, and intermediate seeds were sensitive to low temperature, which could not be stored for a long time through traditional low-temperature seed banks [40]. Cryopreservation was an ideal method for long-term preservation of plant germplasm resources [74]. Cryopreservation was usually carried out in liquid nitrogen at −196 °C. Theoretically, the metabolism and growth activities of plant seeds were in a relatively stable biological state; cells, tissues, and organs did not undergo genetic mutations and trait changes during cryopreservation [75], and various physiological indicators have shown that seed activity was not affected, such as *Phaius tankervilleae* (L’Heritier) Blume seeds [76] and *Tillandsia species* seeds [77]. Cryopreservation safely preserved pollen and young shoots so that their activity was not affected. Pradeep et al. [78] found that the germination percentage of guava pollen was not significantly different from that of fresh state after cryopreservation, thus overcoming the problem of asynchronous flowering. Thomas Rasl reported that the regeneration percentage of winter-acclimated buds of *Dracocephalum austriacum* L. after cryopreservation reached up to 100% [79].

In the process of cryopreservation, the formation of ice crystals and the change of osmotic pressure caused damage and wrinkling of cell membranes, which in turn led to changes in cell and tissue morphology and function. A number of experiments have shown that damage to cell membranes and organelles was strongly related to the formation of intracellular ice crystals [80]. Therefore, some non-orthodox seed crops, such as mango and lychee, were difficult to preserve through cryopreservation. An important reason was that the water content in biological cells was relatively high; freezing occurs during the cooling process, causing damage and destruction to the cell structure, eventually leading to cell death [81]. Due to the sensitivity to dehydration, the water content of non-orthodox seeds was reduced to a safe threshold to survive freezing damage, which was a key factor before the safe cryopreservation [82,83,84].

The germination percentage of *Senna tora* (L.) Roxb. seeds decreased significantly with decreasing the water content with or without liquid nitrogen preservation, and its germination percentage was less than 20% at 9.3% water content [85]. In contrast, the seed germination percentage of *Mimosa pudica* L. seeds was more than 70% when the water content was between 8.09~15.67%, which was non-significantly different before and after liquid nitrogen preservation [85]. Similar to the phenotype of *M*. *pudica* seeds, the germination percentage of *S. tonkinensis* seeds in this study was not significantly different from the control group at 71.2~83% after cryopreservation when the water content was 8.5~10.1% and reached the highest germination rate of 83% at 2 h-dehydration. Since the germination percentage of seeds dehydrated to 5.4% water content was only 30.5%, without preservation value, no cryopreservation treatment was performed. The results of antioxidant enzyme activity identification showed that the SOD activity after cryopreservation reached maximum at 2 h-dehydration, and the difference was not significant before and after cryopreservation. The CAT activity was greatly increased after 2 h-dehydration followed by cryopreservation and reached maximum. The APX activity had a peak at 2 h-dehydration, and there was no significant difference before and after cryopreservation treatment. The physiological processes of dehydration and post-dehydration cryopreservation treatments were similar, both involving protein processing in the ribosome, endoplasmic reticulum, antioxidants, and signal transduction, indicating that cryopreservation treatment had no effect on seeds. In summary, it is concluded that *S. tonkinensis* seeds can be preserved by cryopreservation technology, and the most suitable dehydration time for cryopreservation is 2 h with a water content of 10.1%.

## 4. Materials and Methods

### 4.1. Experimental Materials

The seeds of *S. tonkinensis* used in the experiment were collected in October 2022 from Shishan District, Southwest Guilin, Guangxi (E: 110°16’, N: 25°15). Pick the fresh and mature *S. tonkinensis* seeds together with the pods, peel the seeds from the pods, place the peeled seeds in a cool and dry space, dry them to constant weight, and then put them in the incubator at 5 °C for future use.

The seeds were quickly dehydrated with allochroic silicagel [86]. Seeds with dehydration time of 1, 2, 4, 8, and 24 h were selected and stored in liquid nitrogen for 24 h. Part of seeds were thawed, and the embryos were peeled out; some embryos were fixed with glutaraldehyde fixative for subcellular structure microscopy, and the rest of embryos were ground into powder for protein content, antioxidant enzyme activity determination, and proteomic analysis. The other seeds were subjected to germination experiments.

### 4.2. Germination Experiment

Fresh mature seeds were germinated in incubators at 10 °C, 15 °C, 20 °C, 25 °C, 30 °C, and 35 °C, and the highest germination percentage was found at 25 °C (Appendix A). Fresh mature seeds, seeds dehydrated to various water contents, and seeds cryopreserved after dehydration were used as materials for the germination experiment. Perlite was used as the germination medium, and 50 seeds were sown in 4 replicates for each treatment. The seeds were incubated in an incubator at 25 °C for 30 d (12 h/day alternating light or total darkness, PPFD = 121 µmol m^−2^s^−1^, and the germination state was checked under a green light in total darkness). The survival rate after dehydration treatment and cryopreservation was expressed as the survival rate of seedlings. The data, which were shown as the means ± SDs, were subjected to ANOVA to determine significant differences.

### 4.3. Cell Structure and Antioxidant Enzyme Activity of Seedplumular Axis Treated with Cryopreservation after Dehydration and Dehydration Treatment

The seed plumular axis were washed with water, cut into small cubes of 1 × 1 mm, and placed in the prepared fixative solution (3% glutaraldehyde, 0.1 M PBS, pH 7.0). After rinsing, the phosphate buffer was removed and put into 1% H_2_[OsO_4_(OH)_2_] fixation for 8 h and was then dehydrated with ethanol with a volume fraction gradient of 30~100% for 30 min. After removing the ethanol, infiltration, and embedding, the samples were transferred to epoxy resin for polymerization at 60 °C for 24 h. After the processing, the sample was placed on an ultramicrotome for sectioning with a thickness of 50 nm. After fixing, it was observed and photographed by a transmission electron microscope.

The SOD was measured by the nitro blue tetrazolium (NBT) method [87]. The activity of CAT was measured according to the methods of Maehly and Chance [88], and the activity of APX was measured according to the methods of Hossain and Asada [89]. All assays described above were set up with four biological replicates. The data, which were shown as the means ± SDs, were subjected to *t*-test to determine significant differences. 

### 4.4. Protein Extraction and Enzyme Hydrolysis

Fresh seeds (CK), seeds for 2 h-dehydration (D2), seeds for cryopreservation for 24 h in liquid nitrogen after a 2 h-dehydration (CD2), seeds for 24 h-dehydration (D24), and seeds for cryopreservation for 24 h in liquid nitrogen after 24 h-dehydration (CD24), with three repeat samples in each group, were respectively homogenized to powder in liquid nitrogen. The powder was resuspended in a lysis buffer (7 M urea, 2 M thiourea, 4% SDS, 40 mM Tris-HCl), with a final concentration of 1 mM PMSF and 2 mM EDTA. After mixing, the samples were added to dithiothreitol (DTT) to a final concentration 10 mM then sonicated at 200 W for 15 min. The supernatant was obtained following centrifugation at 13,000× *g* for 20 min at 4 °C and was added to 4 fold volume of ice-cold acetone and precipitated at −20 °C overnight. After centrifugation as described above, the pellets were dissolved in 400 mL, 100 mM TEAB, and 7 M urea. The supernatant was reduced with 10 mM DTT and a water bath for 30 min at 56 °C. Next, the reaction was blocked with 55 mM IAM at dark room temperature for an alkylation reaction lasting 30 min. The homogenates were centrifuged, and the resulting supernatant protein samples were stored at −80 °C. The protein concentration was quantified by Coomassie (Bradford) Protein Assay Kit (Thermo Scientific, Waltham, MA, USA. Bovine albumin (1000 μg/mL) was used as the standard. A total of 100 μg protein from each sample was used for trypsin digestion. Briefly, trypsin was added to protein solution, which was diluted 5 times with 100 mM TEAB at 1:50 and hydrolyzed overnight at 37 °C. The peptides hydrolyzed by the enzyme were desalted by C18 column and then desalinated by vacuum freeze-drying.

### 4.5. iTRAQ Labeling and Grouping

Samples were dissolved with 0.5 M triethylammonium bicarbonate. According to the specification of the iTRAQ-8 marker kit, the samples were marked and mixed. The proteins from each sample were labeled with iTRAQ reagents 113, 114, 115, and 116, respectively. The labeled samples were pooled and purified using a strong cation exchange chromatography column (Phenomenex, Torrance, CA, USA) and separated by liquid chromatography (LC) using the Ultimate 3000 HPLC system, USA). Additionally, the chromatographic column used was Durashell C18 (5 μm, 4.6 × 250 mm). The peptide was separated by increasing an can concentration under alkaline conditions, and the flow rate was 1 mL min^−1^. A total of 42 secondary components were collected and combined into 12 components. The combined components were desalted on the Strata-X column and dried under vacuum. A triple TOF 5600 liquid mass spectrometry system and liquid chromatography-mass spectrometry (LC-MS) were used for mass spectrometry data acquisition. Briefly, the polypeptide sample was dissolved in 2% acetonitrile 0.1% formic acid and analyzed by a Triple TOF 5600 plus mass spectrometer coupled with an Eksigent nano LC system. The polypeptide solution was added to the C18 capture column (5 μm, 100 μm × 20 mm) and eluted at a 90 min time gradient and at a flow rate of 300 nL min^−1^ on the C18 column (3 μm, 75 μm × 150 mm). For IDA (information-dependent collection), the primary mass spectrometry of 30 precursor ions was scanned with an ion accumulation time of 250 ms, and the secondary mass spectrometry of 30 precursor ions was collected with the ion accumulation time of 50 ms. The MS1 spectra were collected in the range of 350 ≤ 1500 m.z., and the MS2 spectra were collected in the range of 100 ≤ 1500 m.z. The dynamic exclusion time of precursor ions was set to 15 s.

In this experiment, peptides were identified using ProteinPilot^TM^ 4.5 (https://en.freedownloadmanager.org/; accessed on 25 April 2022), a search engine matching with AB Sciex 5600 plus (SCIEX, Framingham, MA, USA), with the following paragon parameters. For the identification results of ProteinPilot, we conducted further filtration using unused scores. We chose the peptides with unused scores ≥1.3, which means that the reliability level is more than 95%. The filtrated proteins were used in the subsequent analysis.

### 4.6. Database Search and Proteomic Analysis

Protein identification was performed using ProteomeDiscovererTM 2.2 aligned to the NCBI NR database (http://www.ncbi.nlm.nih.gov/; accessed on 18 January 2022) and SwissProt/UniProt database (http://www.uniprot.org/; accessed on 18 January 2022) by DIAMOND. One protein had to contain at least one unique peptide. Peptide false discovery rate (FDR) analysis ≤0.01 was the filtering parameter. DEPs were identified using a FC (fold change) threshold, and *p*-values of < 0.05 were significant. FC > 1.2 was considered up-regulated, and FC < 0.83 was considered down-regulated.

The COG analysis of the proteins was performed for functional classification via searches in a database (http://www.ncbi.nlm.nih.gov/COG/; accessed on 18 March 2022). Blast2GO 2.5.0 was used for the GO annotations (http://www.geneontology.org/; accessed on 18 March 2022) of the DEPs to classify proteins based on molecular function, biological process, and cellular components. Blast2GO computes Fisher’s Exact Test by applying robust FDR correction for multiple testing and returns a list of significant GO terms ranked by their corrected or one-test *p*-values. KEGG (http://www.genome.jp/kegg/; accessed on 2 April 2022) annotation was performed using KOBAS 2.1.1 software to predict the metabolic pathways. Protein–protein interactions (PPI) were analyzed using the STRING v10 database (http://string-db.org; accessed on 4 February 2022) to determine the identified proteins. 

## 5. Conclusions

In this study, we reported the dehydration characteristics and viability of cryopreservation for *S. tonkinensis* seeds. The results showed that these seeds possessed a certain degree of dehydration tolerance. The highest germination percentage reached 83.0% upon dehydration to a 10.1% water content. However, when the water content dipped below 9.7%, the germination percentage decreased significantly. At a mere 5.4% water content, the germination percentage was only 30.5%. Consequently, the *S. tonkinensis* seeds are best categorized as intermediate seeds. 

Subsequent analysis indicated that the lipid bodies gradually fused during dehydration, and the degree of fusion was significantly improved by cryopreservation. Physiological experiments demonstrated that SOD, POD, and CAT in the antioxidant enzyme system increased during dehydration, aligning with the enrichment tendency of DEPs in proteomics results regarding ROS clearance pathway and stress defense. Proteomics outcomes also unveiled significant DEP enrichment in protein processing in endoplasmic reticulum and ribosomal pathway but not in intracellular energy metabolism and metabolic activities. 

It is postulated that *S. tonkinensis* seeds form an anti-stress protection mechanism mainly by protein processing in the endoplasmic reticulum and the antioxidant system during dehydration, which is related to their dehydration tolerance. In addition, DEPs were significantly enriched in the ribosome pathway; only three differentially expressed LEA proteins were identified, and it is speculated that the lack of specific LEA and dehydrins is also one of the important factors for the reduced germination percentage after excessive dehydration. There was no significant difference in protein enrichment pathway between dehydrated seeds and 2 h-dehydrated followed by cryopreservation, suggesting that the cryopreservation of *S. tonkinensis* seeds is indeed feasible.

## Figures and Tables

**Figure 1 plants-12-01842-f001:**
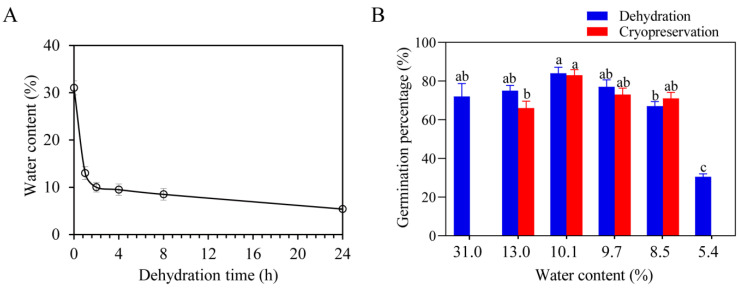
Dynamic map of water content and germination percentage of *S. tonkinensis* seeds. (**A**) Changes of seed water content with dehydration time. (**B**) Changes of seed germination percentage with water content before and after cryopreservation treatment. Note: Different letters on top of each bar indicate statistically significant differences (*p* < 0.05).

**Figure 2 plants-12-01842-f002:**
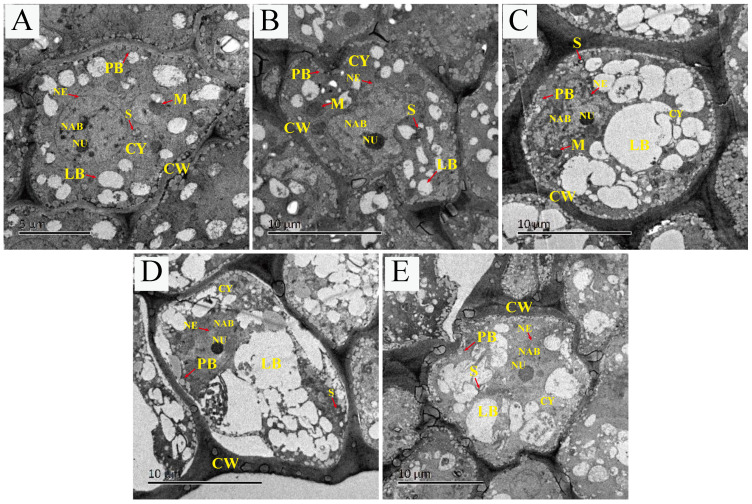
Subcellular structural changes of plumular axis of *S. tonkinensis* seeds after dehydration and dehydration followed by cryopreservation. (**A**) Control. (**B**) Dehydration for 2 h. (**C**) Cryopreservation for 24 h in liquid nitrogen after 2 h-dehydration. (**D**) Dehydration for 24 h. (**E**) Cryopreservation for 24 h in liquid nitrogen after 24 h-dehydration. NU, nucleus; CW, cell wall; LB, lipid bodies; M, mitochondria; PB, protein bodies; S, starch grain; CY, cytoplasm; NE, nuclear envelope, NAB, nucleolus associated bodies.

**Figure 3 plants-12-01842-f003:**
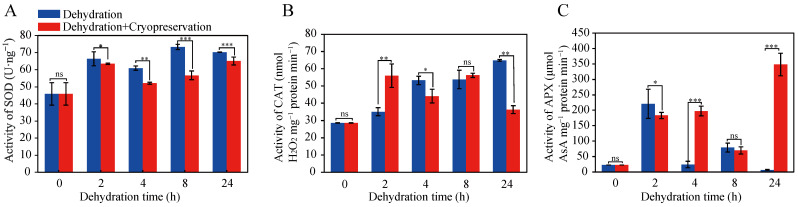
Determination of antioxidant enzyme activity after dehydration and dehydration followed cryopreservation in *S. tonkinensis* seed plumular axis. (**A**) Changes of SOD activity with dehydration time before and after cryopreservation. (**B**) Changes of CAT activity with dehydration time before and after cryopreservation. (**C**) Changes of APX activity with dehydration time before and after cryopreservation. Note: *, **, and *** mean that the difference between the two groups is less than 0.05 (*p* < 0.05), 0.01 (*p* < 0.01), 0.001 (*p* < 0.001), respectively; ns means that the difference between the two groups is more than 0.05.

**Figure 4 plants-12-01842-f004:**
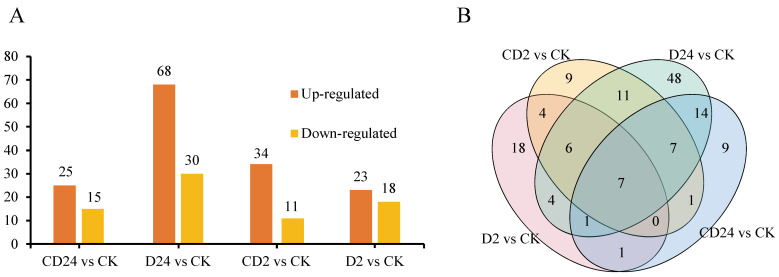
The profile of the differentially expressed proteins (DEPs) in comparison groups (D2/CK, CD2/CK, D24/CK, and CD24/CK). (**A**) The number of DEPs identified in comparison groups. (**B**) Venn diagram analysis of DEPs among different comparison groups.

**Figure 5 plants-12-01842-f005:**
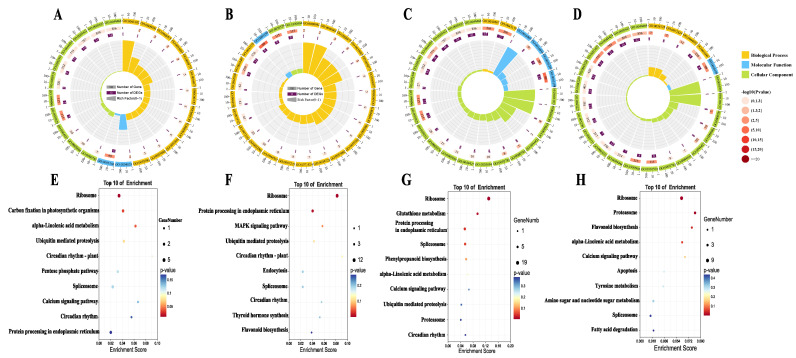
GO and KEGG enrichment results of DEPs. (**A**–**D**) GO enrichment analysis of DEPs in D2 vs. CK, CD2 vs. CK, D24 vs. CK, and CD24 vs. CK. (**E**–**H**) KEGG enrichment analysis of DEPs in D2 vs. CK, CD2 vs. CK, D24 vs. CK, and CD24 vs. CK.

**Figure 6 plants-12-01842-f006:**
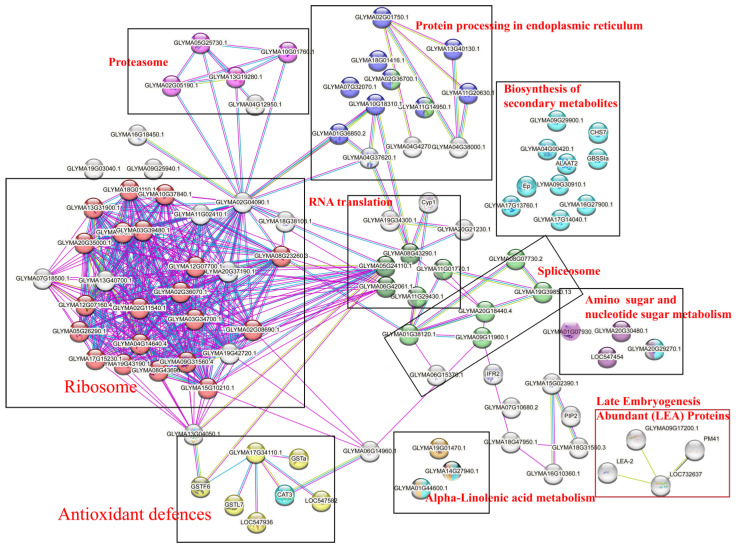
Interaction network of DEPs.

## Data Availability

The data is contained within the manuscript and Appendix A.

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
