# Peer review of "iTRAQ-Based Proteomic and Physiological Analyses Reveal the Mechanisms of Dehydration and Cryopreservation Tolerance of *Sophora tonkinensis* Gagnep. Seeds"

_plants, 2023, doi:10.3390/plants12091842_

Round 1
Reviewer 1 Report
The manuscript is valuable and informative. The Introduction is well-prepared, Results are well-presented, and the Discussion is profound. I only have some technical suggestions concerning the manuscript, which should be considered before publishing the text.
- The authors are using an outdated name of the family Fabaceae.
- Some parts of the text are unclear (please, see the corrected manuscript).
- Keywords should be arranged alphabetically and not repeating words from the title.
- References 2-4 are old and require an update.
- Latin names of species and genera should be written in italics, including the Reference list.
- MDPI uses serial coma.
- The caption to Figure 2 should be more descriptive. See DOI: 10.1016/j.scienta.2017.04.019
- Some parts of the Results should be transferred either to the Materials and methods section (although, avoid repetitions) or Discussion sections.
- The first paragraph of the Discussion is a repetition of existing knowledge and is a repetition of the Introduction.
- Please, provide the initials of the authors who first described the species (when first mentioned).
- The English form is generally good, although some minor corrections are required.
- Sometimes, spaces between words are missing.
- Line 473: correct chemical symbols.
- In the Materials and methods section, the number of repetitions and replications is unclear.
- Units should be presented as a product.
- What protein was used as the standard in the biochemical array?
- In the reference list, the words of some articles are all capitalized and some are not. Please follow the MDPI formatting style. Moreover, journal names should be abbreviated.
For more specific comments, please see the corrected manuscript.

Author Response
Response to Reviewer 1 Comments
The manuscript is valuable and informative. The Introduction is well-prepared, Results are well-presented, and the Discussion is profound. I only have some technical suggestions concerning the manuscript, which should be considered before publishing the text.
Response: We appreciate your consideration of our manuscript.
Point 1: The authors are using an outdated name of the family Fabaceae.
Response 1: Thanks for your correction. The name of the family has been modified to Fabaceae, please see line 24.
Point 2: Some parts of the text are unclear (please, see the corrected manuscript).
Response 2: We have corrected the issue you pointed out, please see the corrected manuscript.
Point 3: Keywords should be arranged alphabetically and not repeating words from the title.
Response 3: The keywords are arranged alphabetically, and no words from the title are repeated, please see line 44.
Point 4: References 2-4 are old and require an update.
Response 4: We have updated References 2-4, please see lines 583-588.
Point 5: Latin names of species and genera should be written in italics, including the Reference list.
Response 5: We have italicized all the Latin names of species and genera, please see the corrected manuscript.
Point 6: MDPI uses serial coma.
Response 6: Serial commas are used in line 84.
Point 7: The caption to Figure 2 should be more descriptive. See DOI: 10.1016/j.scienta.2017.04.019
Response 7: We have modified the caption in figure 2 based on your suggestion,please see lines 163-165.
Point 8: Some parts of the Results should be transferred either to the Materials and methods section (although, avoid repetitions) or Discussion sections.
Response 8: We have reviewed the Results of our manuscript and made the necessary transfers. Thank you again for your valuable feedback, please see the corrected manuscript.
Point 9: The first paragraph of the is a repetition of existing knowledge and is a repetition of the Introduction.
Response 9: We have addressed the comments by modifying the content of Discussion 3.1. Specifically, we removed the previous version of 3.1 and merged the relevant content with Discussion 3.2 to eliminate redundancy.
Point 10: Please, provide the initials of the authors who first described the species (when first mentioned).
Response 10: The initials of the authors who first described the species have been provided, please see the corrected manuscript.
Point 11: The English form is generally good, although some minor corrections are required.
Response 11: We have carefully checked and improved the English writing in the revised manuscript.
Point 12: Sometimes, spaces between words are missing.
Response 12: We have thoroughly reviewed the text and made sure all relevant formats are corrected.
Point 13: Line 473: correct chemical symbols.
Response 13: The chemical symbols has been corrected, please see line 461.
Point 14: In the Materials and methods section, the number of repetitions and replications is unclear.
Response 14: Four biological repetitions were conducted in this study. The sentence has been rewritten in lines 470-471.
Point 15: Units should be presented as a product.
Response 15: The units have been corrected, please see line 453.
Point 16: In the reference list, the words of some articles are all capitalized and some are not. Please follow the MDPI formatting style. Moreover, journal names should be abbreviated.
Response 16:We have adhered to the MDPI formatting style for the references in our manuscript, including the proper abbreviation of journal names.
Point 17: What protein was used as the standard in the biochemical array?
Response 17: Bovine albumin (1000 μg/ml) was used as the standard in the biochemical array.
Point 18: For more specific comments, please see the corrected manuscript.
Response 18: We appreciate your patience and careful review of our work. We have thoroughly addressed all the deficiencies based on the corrected manuscript.

Reviewer 2 Report
In this study, the authors aimed to investigate the response mechanism of S. tonkinensis seeds to dehydration using different experimental approaches, mainly proteomic and metabolomic, to explore the effect of cryopreservation treatment of seeds.
The work is interesting and adds new information to the field. The manuscript provides theoretical aspects for the preservation of the resources of intermediate seeds. The research is interesting and rigorous. The state of the art is well documented and focuses on the topic precisely.
The aims are clearly stated.
The methodologies adopted and the general experimental plan appear to be adequate. All of them are described in detail and the general experimental is definitively appropriate to achieve the aims.
The results are described in detail and well documented. The authors provided a large set of data, including supplementary materials.
The discussion and conclusions are well addressed
I recommend that the authors recheck the manuscript for redundancies and grammatical errors.
Author Response
Response to Reviewer 2 Comments
In this study, the authors aimed to investigate the response mechanism of S. tonkinensis seeds to dehydration using different experimental approaches, mainly proteomic and metabolomic, to explore the effect of cryopreservation treatment of seeds.
The work is interesting and adds new information to the field. The manuscript provides theoretical aspects for the preservation of the resources of intermediate seeds. The research is interesting and rigorous. The state of the art is well documented and focuses on the topic precisely.
The aims are clearly stated.
The methodologies adopted and the general experimental plan appear to be adequate. All of them are described in detail and the general experimental is definitively appropriate to achieve the aims.
The results are described in detail and well documented. The authors provided a large set of data, including supplementary materials.
The discussion and conclusions are well addressed
I recommend that the authors recheck the manuscript for redundancies and grammatical errors.
Response: We appreciate your consideration of our manuscript. We have carefully checked the entire manuscript and made significant improvements to eliminate redundancies and correct grammatical errors. We believe that the quality of the revised version now meets the standards for publication.
Reviewer 3 Report
The manuscript entitled “iTRAQ-based proteomic and physiological analyses reveal the mechanisms of dehydration and cryopreservation tolerance of Sophora tonkinensis seeds” by Luo. et al., describes the morphological, physiological, biochemical and protein expression characteristics of S. tonkinensis seeds treated with dehydration and cryopreservation treatment through cell microstructure observation, antioxidant enzyme activity assay and proteomic analysis, respectively. This study is systematic and meaningful.
However, the major problem for the study is the writing and data analysis. For example, the aim of this study is not clear. In the Abstract, line 26, the authors wrote the purpose is to investigate the physiological mechanism of seed dehydration tolerance, but why select cryopreservation treatment? Why you chose the iTRAQ-based proteomics technique to reveal the mechanism?
Addtionally, the results of proteomics were too descriptive, in other words, it’s too much description related to the profile of DEPs. That’s why in part 3.2 molecular mechanism of dehydration tolerance, the authors only cited table S2. The current figs and tables can not contribute to the explanation of mechanism. So, I suggest the authors to go through the proteomics data and find out the potential regulatory mechanism to prepare the new fig 7 using proteomics data.
For the discussion writing, besides the references were used, the current result needs to be cited to be discussed. So, I suggest authors to go back to their resutls / data, to do the more specific analysis instead of profile analysis.
Fig 1B, 31%water content, there is no data for cryopreservation, does it mean the gemination percentage for 31% is zero? The same thing for 5.4% water content. And I cann’t understand the statistical analysis in fig 4B, please clarify the statistic method in M & M section. It seems one-way ANOVA is used among all the groups because the different letters was used. However, it seems t-test is used between two groups.
Fig 3C, 24 h, why there is the letter b? Fig 4A, there is no line in y-axis. Fig 4B legend, line 213, number of identified DEPs.
Figure 5, see the right side, some information is out of the pdf.
Figure 6, “Agains oxidative damage”? Against?
For the protein identification, how many peptides or unique peptides was used?
These are just examples, please check the whole manuscript with the mistakes.
Author Response
Response to Reviewer 3 Comments
The manuscript entitled “iTRAQ-based proteomic and physiological analyses reveal the mechanisms of dehydration and cryopreservation tolerance of Sophora tonkinensis seeds” by Luo. et al., describes the morphological, physiological, biochemical and protein expression characteristics of S. tonkinensis seeds treated with dehydration and cryopreservation treatment through cell microstructure observation, antioxidant enzyme activity assay and proteomic analysis, respectively. This study is systematic and meaningful.
Point 1: However, the major problem for the study is the writing and data analysis. For example, the aim of this study is not clear. In the Abstract, line 26, the authors wrote the purpose is to investigate the physiological mechanism of seed dehydration tolerance, but why select cryopreservation treatment?
Response 1: Thank you for your comment. We have revised the abstract to better reflect the purpose of our article. Our study aims to investigate the physiological mechanisms underlying seed dehydration tolerance and to identify optimal preservation methods. Given the challenges associated with long-term preservation of recalcitrant seeds, we have focused on exploring cryopreservation as a promising research direction [72, 73]. Accordingly, we have specifically chosen cryopreservation as the preservation method to investigate in our study (Lines 26-27 and Discussion 3.2). We hope that this updated abstract provides a clearer overview of our research objectives.
Point 2: Why you chose the iTRAQ-based proteomics technique to reveal the mechanism?
Response 2: iTRAQ is a well-established and potent technique utilized in quantitative proteomics with low systematic error and high quantitative accuracy. It enables the simultaneous comparison of up to 8 different samples in a single experiment, thus reducing experimental variation, conserving time and resources, and enhancing statistical power. Hence, it is an appropriate choice for our experimental design. By comparing the proteome expression profiles of germplasm before and after dehydration and cryopreservation, the molecular mechanism of germplasm response to dehydration and ultra-low temperature preservation can be elucidated at the protein expression level.
Point 3: Addtionally, the results of proteomics were too descriptive, in other words, it’s too much description related to the profile of DEPs. That’s why in part 3.2 molecular mechanism of dehydration tolerance, the authors only cited table S2. The current figs and tables can not contribute to the explanation of mechanism. So, I suggest the authors to go through the proteomics data and find out the potential regulatory mechanism to prepare the new fig 7 using proteomics data.For the discussion writing, besides the references were used, the current result needs to be cited to be discussed. So, I suggest authors to go back to their resutls / data, to do the more specific analysis instead of profile analysis.For the discussion writing, besides the references were used, the current result needs to be cited to be discussed. So, I suggest authors to go back to their resutls / data, to do the more specific analysis instead of profile analysis.
Response 3:Thank you very much for the suggestions. In Discussion 3.2(now 3.1), we not only cited Table S2, but also referenced all relevant proteomic results (such as Figures 5 and 6). We also made sure to annotate the referenced chart after discussion, as indicated in line 344-347.
In Discussion 3.1 (previous 3.2), we cited some current results for discussion, as shown in lines 309-310, 343-350.
Furthermore, we reorganized the regulatory mechanism to improve clarity and coherence in the manuscript, although we did not redo Figure 7, we believe that the revisions have strengthened the manuscript overall.
Point 4: Fig 1B, 31%water content, there is no data for cryopreservation, does it mean the gemination percentage for 31% is zero? The same thing for 5.4% water content. And I cann’t understand the statistical analysis in fig 4B, please clarify the statistic method in M & M section. It seems one-way ANOVA is used among all the groups because the different letters was used. However, it seems t-test is used between two groups.
Response 4:
Fresh harvested seeds (with a water content of 31.0%) all died after cryopreservation. Please see lines 119-120.
The germination percentage of seeds dehydrated to 5.4% water content was only 30.5%, which has no value for cryopreservation. Therefore, there was no data for cryopreservation. Please see lines 419-421.
Figure 4B is a Ven plot aimed at exploring the number of common DEPs of the four comparision groups.
The statistical methods used in Figure 3 and Figure 1 are indeed different. Specifically, Figure 1 aims to explore the differences in seed germination percentage with different water contents and test whether there are significant differences between multiple sets of data. Therefore, we chose to use one-way ANOVA. In contrast, Figure 3 compares the enzyme activity differences of seeds with the same water content before and after low-temperature treatment to test whether there is a significant difference between the two sets of data. For this purpose, we chose to use a t-test.
Point 5: Fig 3C, 24 h, why there is the letter b? Fig 4A, there is no line in y-axis. Fig 4B legend, line 213, number of identified DEPs.
Response 5: Due to formatting errors when exporting images, the letter b appearing in Figure 3C has been modified and statistical methods have been added to M&M.
Figure 4A shows the bar chart of the number of DEPs in comparison groups (D2/CK, CD2 /CK, D24/CK and CD24/CK). We have replotted the chart and added the chart notes. The legend of Fig 4B has been modified to “Veen diagram analysis of DEPs among different comparison groups”.
Point 6: Figure 5, see the right side, some information is out of the pdf.
Response 6: We have resized the image to ensure that it is within the field of view of the manuscript. Please see Figure 5.
Point 7: Figure 6, “Agains oxidative damage”? Against?
Response 7: Thank you for your attention. We have changed the expression to “antioxidant defence”. Please see Figure 6.
Point 8: For the protein identification, how many peptides or unique peptides was used?
Response 8: In this study, a total of 978,057 spectra were obtained, and 12,530 peptides, 9568 unique peptides and 3898 proteins were identified. Relevant information is added in Figure S1.
Point 9: These are just examples, please check the whole manuscript with the mistakes.
Response 9: Thank you for your comments. The manuscript has undergone a thorough review, and necessary corrections have been made. Consequently, we are confident that the enhanced quality of the manuscript meets the publication standards.

Round 2
Reviewer 3 Report
1. Fig1B, you should check the statistical analysis. Because please see dehydration group 8.5% and 5.4% data, the letter is c which means there is no significance, however, with our eyes, the histogram between two data is largely different. Another example is cryopreservation group of 13% and dehydration group of 8.5%, the letter is b and c respectively (means they were significance). However, with our eyes, we can see the two histogram is almost the same, are you sure it's significance? So, I suggest the authors to check your data.
Additionally, for a pre-published fig, all the symbol in figs, should be explained in the legend, in fig1B, there is no explaination of different letters. However, it's the format problem, I mean the authors should improve the figs or tables with a high standard.
2. For point 8, I mean how many peptides / unique peptides was used for each protein identification. In other words, the criteria for the protein identification in the current study. For example, the identified proteins with less than one unique pepetides were removed in the literature. That is one of the criteria for protein identification.
Author Response
Comments and Suggestions for Authors
Point 1:Fig1B, you should check the statistical analysis. Because please see dehydration group 8.5% and 5.4% data, the letter is c which means there is no significance, however, with our eyes, the histogram between two data is largely different. Another example is cryopreservation group of 13% and dehydration group of 8.5%, the letter is b and c respectively (means they were significance). However, with our eyes, we can see the two histogram is almost the same, are you sure it's significance? So, I suggest the authors to check your data. Additionally, for a pre-published fig, all the symbol in figs, should be explained in the legend, in fig1B, there is no explaination of different letters. However, it's the format problem, I mean the authors should improve the figs or tables with a high standard.
Response: We appreciate your careful examination of our work. Upon reevaluating our data, we realized that the issue might stem from an error in our plotting script. We have thoroughly rechecked the statistical tests and adjusted the labeling of significance in the figure accordingly. We believe that the updated figure and analysis will provide a clearer and more accurate representation of the results.
Point 2:For point 8, I mean how many peptides / unique peptides was used for each protein identification. In other words, the criteria for the protein identification in the current study. For example, the identified proteins with less than one unique pepetides were removed in the literature. That is one of the criteria for protein identification.
Response: A minimum of one unique peptide per protein was required for identification. We chose this threshold to balance the need for confident protein identification while maximizing the number of proteins identified in our dataset. It is important to note that Sophora tonkinensis does not have a sequenced genome available. Therefore, our approach aimed to maximize proteome coverage while acknowledging the limitations imposed by the lack of a reference genome. Although this approach may result in a slightly higher risk of false-positive identifications, we believe that the benefits of increased proteome coverage outweigh this risk. We have made the clarification in the Method section , See line 529.